# SinkQ: Accurate KV Cache Quantization with Dynamic Sink Tracking

## Abstract

The impressive capabilities of large language models (LLMs) come at the cost of substantial computational resources during deployment. While KV Cache can significantly reduce recomputation during inference, it also introduces additional memory overhead. KV Cache quantization presents a promising solution, striking a good balance between memory usage and accuracy. Previous research has shown that the Keys are distributed by channel, while the Values are distributed by token. Consequently, the common practice is to apply channel-wise quantization to the Keys and token-wise quantization to the Values. However, our further investigation reveals that a small subset of unusual tokens exhibit unique characteristics that deviate from this pattern, which can substantially impact quantization accuracy. Furthermore, these tokens often have higher attention scores, exacerbating their quantization errors. To address this, we develop a simple yet effective method to identify these tokens accurately during the decoding process and exclude them from quantization, significantly improving overall accuracy. Extensive experiments show that our method achieves significant accuracy improvements under 2-bit quantization and can deliver a 6.4× reduction in memory usage and a 2.3× increase in throughput. Our code will be released upon acceptance.

## 1 Introduction

Large language models (LLMs) have significantly impacted various industries due to their powerful capabilities (Achiam et al., 2023; Touvron et al., 2023a;b; Dubey et al., 2024; Jiang et al., 2023). However, their auto-regressive nature makes the generation process slow. While using KV Cache can reduce decoding complexity from $O(n^2)$ to $O(n)$ by storing the Keys and the Values computed during inference, it introduces substantial memory overhead. This overhead scales with sequence length, batch size, and hidden dim, often creating a memory bottleneck and placing considerable pressure on resources during deployment. As a result, optimizing KV Cache management to enhance resource utilization and improve model throughput remains a critical challenge.

KV Cache affects throughput in two primary ways. First, its memory usage limits the scalability of batch sizes, reducing parallelism during decoding and thus lowering throughput. Second, attention computation is delayed while waiting for the KV Cache to be transferred from memory to the computation unit. As the KV Cache size grows, the transmission time increases, decreasing throughput. Existing approaches mainly address this issue by optimizing hardware scheduling (Aminabadi et al., 2022; Dao et al., 2022; Sheng et al., 2023; Kwon et al., 2023) and reducing the size of the KV Cache (Liu et al., 2024b; Hooper et al., 2024; Kang et al., 2024; Zhang et al., 2023; Xiao et al., 2024). In this paper, we focus on the latter approach—KV Cache compression.

One method to reduce the size of the KV Cache is to reduce the number of values that need to be stored. The shape of the KV Cache is $[num\_layers, batch\_size, num\_heads, sequence\_length, head\_dim]$. There are various compressing methods across each dimension, including layer-wise KV Cache sharing (Wu & Tu, 2024; Brandon et al., 2024; Zuhri et al., 2024; Mu et al., 2024), prefix sharing (Juravsky et al., 2024; Zhu et al., 2024), head-wise KV Cache sharing (Shazeer, 2019; Ainslie et al., 2023), token eviction (Xiao et al., 2024; Zhang et al., 2023; Ge et al.), and low-rank projection (Wang et al., 2024; Yu et al., 2024; Chang et al., 2024; Liu et al., 2024a).

Another strategy for reducing the size of KV Cache is quantization. However, unlike weight quantization, KV Cache quantization poses unique challenges due to the uneven distribution of the Keys

and Values (Kang et al., 2024). To enhance quantization accuracy, various methods have been proposed, including using low-rank matrices to approximate the error before and after quantization (Kang et al., 2024), smoothing Key distributions through specific mappings (Ashkboos et al., 2024; Chang et al., 2024), channel-wise Key and token-wise Value asymmetric quantization (Liu et al., 2024b; Hooper et al., 2024), non-uniform quantization (Hooper et al., 2024; Dettmers et al., 2022), mixed-precision quantization (Dong et al., 2024), and Block Floating Point (BFP) quantization (Trukhanov & Soloveychik, 2024). Among these methods, channel-wise Key and token-wise Value asymmetric quantization has garnered much attention for its high accuracy and tuning-free nature. This technique operates under the assumption that some channels of the Keys have huge magnitudes and that the distribution of the Keys within the same channel is relatively uniform.

However, our further exploration reveals that a few unusual tokens deviate from this assumption. Moreover, we find that these tokens exhibit remarkably high attention scores—often referred to as attention sinks (Xiao et al., 2024). Notably, unlike previous studies, we find that these attention sinks can occur at any position within a sentence rather than being confined to the initial positions. Based on these observations, we propose Sink-aware KV Cache Quantization (*SinkQ*), a simple yet effective method that identifies these tokens and excludes them from the quantization process, thereby improving quantization accuracy. With hardware-friendly implementation, *SinkQ* achieves significant accuracy improvements under 2-bit quantization, resulting in a 6.4× reduction in memory usage and a 2.3× increase in throughput.

Overall, our contributions are summarized as follows:

- We investigate the outlier channels of the KV Cache and identify that some tokens deviate from the previous assumptions.
- We introduce Sink-aware KV Cache Quantization (*SinkQ*), a simple yet effective method to dynamically identify and exclude these tokens during quantization, thus improving overall quantization accuracy.
- Our method achieves significant accuracy improvements under 2-bit quantization, yielding a 6.4× reduction in memory usage and a 2.3× increase in throughput, thereby enhancing model efficiency.

## 2 BACKGROUND

**Implementation of KV Cache.** Transformer-based (Vaswani, 2017) LLMs typically utilize KV cache to prevent the redundant calculation of the attention scores and accelerate auto-regressive decoding. The generation process of LLMs with KV cache is divided into the prefill phase and the decoding phase (Patel et al., 2024). Given a prompt $X = \{x_0, x_1, \ldots, x_{n-1}\}$ and tensor $\boldsymbol{X} \in \mathbb{R}^{b \times n \times d}$ after embedding, where $b$ is the batch size, $n$ is the length of the prompt, and $d$ represents the hidden size, we will briefly describe the calculation process of the attention block, and we omit the number of heads in the multi-head attention mechanism.

*i)* During the prefill phase, the Keys $\boldsymbol{K}_{<n}$ and Values $\boldsymbol{V}_{<n}$ are computed and cached by transforming $\boldsymbol{X}$ through the Key and Value weight matrices $\mathbf{W}_k, \mathbf{W}_v \in \mathbb{R}^{d \times d}$ of each layer, which can be formulated as:
$$\boldsymbol{K}_{<n} = \boldsymbol{X}\mathbf{W}_k, \quad \boldsymbol{V}_{<n} = \boldsymbol{X}\mathbf{W}_v.$$

*ii)* During the decoding phase, only the Keys and Values of the new token $x_n$ need to be calculated, which are then combined with the cached Keys and Values to compute the new attention scores and outputs. For the current input tensor $\boldsymbol{X}_n \in \mathbb{R}^{b \times 1 \times d}$, we update the KV cache as follows:
$$\boldsymbol{K} = \boldsymbol{K}_{<n} \| \boldsymbol{K}_n, \quad \boldsymbol{V} = \boldsymbol{V}_{<n} \| \boldsymbol{V}_n,$$

where $\boldsymbol{K}_n = \boldsymbol{X}_n\mathbf{W}_k$ and $\boldsymbol{V}_n = \boldsymbol{X}_n\mathbf{W}_v$. We calculate the new attention output ATT as follows:

$$\boldsymbol{Q}_n = \boldsymbol{X}_n\mathbf{W}_q, \quad \text{ATT} = \text{Softmax}\left(\frac{\boldsymbol{Q}_n\boldsymbol{K}^{\top}}{\sqrt{d_k}}\right)\mathbf{V}, \quad (1)$$

where $\mathbf{W}_q$ is the query weight matrix in the corresponding layer and $\sqrt{d_k}$ is the normalization factor.

**Necessity of compression.** While KV cache reduces the computational complexity from $O(n^2)$ to $O(n)$, it introduces substantial GPU memory overhead, particularly with long sequence lengths and

large batch sizes. For example, in the case of LLaMA3-8B (Dubey et al., 2024), where the number of layers $n_{\text{layers}}$ is 32, the number of heads $h$ is 8, the head dimension $d$ is 512, the input length $l$ is 8192, and the batch size $b$ is 64, performing inference with fp16 precision (which uses 2 bytes per value) requires $4bhdln_{\text{layers}}$ bytes to store the KV cache—equivalent to 256GB of memory. Thus, effectively compressing the KV cache is crucial to reducing GPU memory usage.

**Uniform Quantization.** In this paper, we focus on compressing the KV cache by reducing the bit-width needed to represent cached tensors. A straightforward approach is Uniform Quantization (Jacob et al., 2018), which maps continuous numerical data to a discrete domain. Specifically, to quantize a high-precision matrix (e.g., float32) $\boldsymbol{X}$ to a matrix $\boldsymbol{X}'$ with $b$-bit precision, we first determine the quantization step size $q$. Each element $X_{i,j} \in \boldsymbol{X}$ can then be quantized to $Q(X_{i,j})$ as follows:

$$Q(X_{i,j}) = \lfloor (X_{i,j} - \boldsymbol{X}_{min})/q \rceil, \; q = (\boldsymbol{X}_{max} - \boldsymbol{X}_{min})/(2^b - 1), \tag{2}$$

where $\lfloor \cdot \rceil$ is the rounding function.

**Group Quantization.** However, Uniform Quantization does not fully exploit the distribution characteristics of the data, which can lead to significant quantization errors, especially when there are outliers. A more advanced technique is Group Quantization (Yao et al., 2022), which divides the matrix into multiple groups, expecting the data within each group to share similar distribution characteristics. Unlike Uniform Quantization, Group Quantization allows each group to have different quantization parameters, such as step size. This flexibility enables the method to better adapt to the local characteristics of the data, thereby reducing quantization errors while maintaining a low bit-width. The channel-wise Key quantization and token-wise Value quantization proposed by KIVI (Liu et al., 2024b) is a type of Group Quantization.

## 3 METHOD

In this section, we propose Sink-Aware KV Cache Quantization (*SinkQ*). We start with a preliminary exploration of the Keys and Values before introducing our method.

### 3.1 EXPLORATION OF THE KEYS AND VALUES.

We conduct a series of preliminary experiments to gain a deeper understanding of the Keys and Values. For illustration, we take a sentence generated by LLaMA-2-7b-chat-hf [1] as an example. Table 5 in the Appendix presents the prompt and the generated context.

**Distribution of the Keys and Values.** Figure 1a and Figure 1d display the magnitude of the Keys and Values from layer 10, head 17. Notably, some channels exhibit exceptionally large Keys, and within these channels, the distribution of the Keys appears relatively uniform. In contrast, the Values have no distinct characteristics. These observations are consistent with those reported in KIVI (Liu et al., 2024b).

**Distribution in outlier channels.** We further investigate the distribution of these outlier channels. Figure 1b shows the Keys in an outlier channel from layer 10, head 17 (we plot the first 100 tokens). While the Keys generally exhibit a uniform distribution, a few tokens are notable exceptions. This pattern becomes clearer after sorting, as shown in Figure 1e, where some Keys have very small values while others are significantly larger. These exceptions can substantially increase $\boldsymbol{X}_{max} - \boldsymbol{X}_{min}$ in Equation 2 during quantization, ultimately diminishing quantization accuracy.

**Identifying Outlier Tokens.** Intuitively, tokens with very small magnitude of the Keys in outlier channels are also likely to have smaller magnitude overall. To test this hypothesis, we plot the Keys from an outlier channel and the magnitude of the Keys across all channels (we plot the first 300 tokens). As shown in Figure 4 in the Appendix, the results confirm our assumption, suggesting that we can efficiently and accurately identify these outlier tokens with the magnitude of the Keys.

**Outlier Tokens & Attention scores.** We further explore the attention scores of the outlier tokens and find that these tokens tend to have exceptionally high attention scores. To visualize this, we compare the overlap between the N tokens with the smallest magnitude of the Keys and the N tokens with the highest attention scores (we plot the first 300 tokens). Figure 1c presents our results.

---

[1] https://huggingface.co/meta-llama/Llama-2-7b-chat-hf

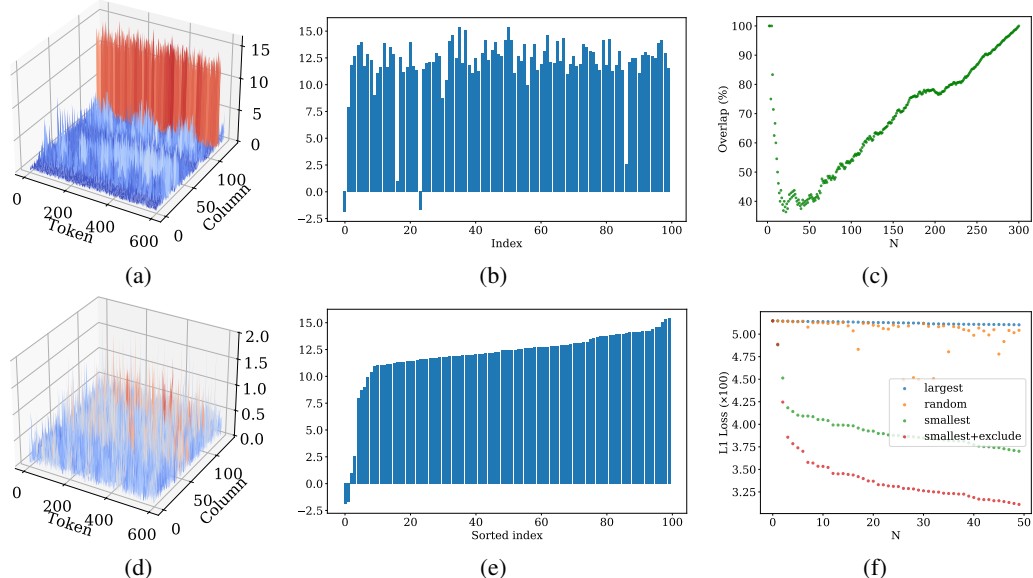

(a) (b) (c)

(d) (e) (f)

Figure 1: Observations from preliminary experiments: (a) The Keys are distributed by channel and have some outlier channels. (d) The distribution of the Values does not exhibit any notable characteristics. (b) In certain outlier channels, a few tokens with low magnitude of Keys disrupt the originally uniform distribution within these channels. (e) Visualization of the sorted Keys in an outlier channel shows a rapid increase from a low value to very high values. (c) There is a significant overlap between the smallest N Keys and the largest N attention scores when N is small. (f) The L1 loss of attention output before and after quantization by retaining full-precision tokens based on different criteria. The best result is retaining full-precision tokens with the smallest magnitude of the Keys and excluding these tokens during quantization.

When N is small, the overlap is initially high but quickly diminishes to a minimum before gradually increasing again. Notably, the tokens selected when N is small correspond to the outlier tokens, indicating that outlier tokens generally have high attention scores.

**Removing Outlier Tokens.** From our analysis, outlier tokens significantly impact the effectiveness of quantization. These outlier tokens typically exhibit high attention scores, indicating that even minor quantization errors can result in considerable losses in attention output. By retaining these outlier tokens with full precision, we can greatly reduce the loss of attention output. To investigate this further, we retain different numbers of full-precision tokens based on different selection criteria and compare the L1 loss of attention outputs before and after quantization. The results (Figure 1f) reveal that retaining tokens with the largest keys yields the worst performance, while retaining those with the smallest Keys provides the best results, aligning with our previous findings. Furthermore, outlier tokens contribute to an increase in $X_{max} - X_{min}$ within outlier channels, thus affecting quantization accuracy. To address this issue, we exclude these outlier tokens during quantization, leading to even better results.

### 3.2 *SinkQ*: Sink-aware KV Cache Quantization

From these observations, we note that a few outlier tokens exhibit exceptionally high attention scores, consistent with the findings from StreamingLLM (Xiao et al., 2024). These tokens are referred to as attention sinks, and we will use this term later. Previous work suggesting that attention sinks occur only in the initial tokens; however, our observations reveal that they can appear at any position within a sentence. Moreover, the distribution of the Keys of attention sinks differs significantly from that of other tokens, which can substantially impact overall quantization performance.

Based on the insights above, we propose Sink-aware KV Cache Quantization (*SinkQ*). Figure 2 illustrates an overview of our method. *SinkQ* consists of two components: quantization and decoding.

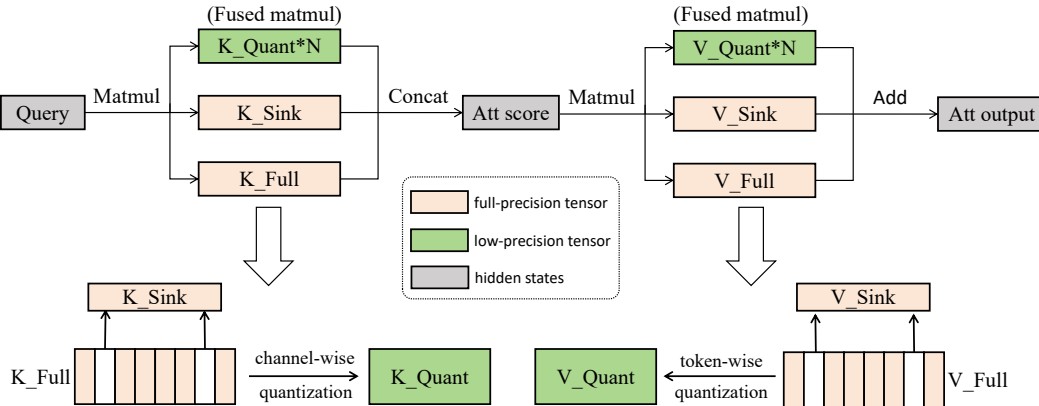

Figure 2: Overview of *SinkQ*. Top: Decoding stage. Multiply the Query by each type of the Keys and concatenate the results to obtain the attention scores. Multiply the attention scores by each type of the Values and sum the results to get the attention output. Bottom: Quantization stage. Before quantization, process the sinks first, then quantize the Keys by channel and the Values by token.

**Quantization** We define a fixed-size sink pool with a capacity of $sink\_num$ to store the Keys and Values of the attention sinks. To fulfill group quantization, we quantize KV Cache every $G$ steps, which means group size, a hyper-parameter in group quantization. When the group is full, tokens selected for quantization will obtain a position in the sink pool based on the rules described in Section 3.1. Once selected, the Keys and Values of these tokens are replaced with the mean values of all tokens to eliminate their impact on quantization. We adopt the quantization strategy from KIVI, applying channel-wise quantization for the Keys and token-wise quantization for the Values.

**Decoding** We maintain three types of KV Cache: the quantized KV Cache, the full-precision KV Cache, and the KV Cache stored in the sink pool. First, the Query is multiplied by all three types of Keys, and we concatenate the results to produce the attention scores. Next, we multiply these scores by their corresponding Values from each type and sum them to generate the final attention output. To enhance decoding efficiency, we utilize a CUDA fused kernel to multiply full-precision and quantized matrices efficiently.

Following KIVI, we have group tokens and recent tokens. We trigger a new quantization step when the full-precision KV Cache reaches a certain length (necessary for channel-wise key quantization). When the group is not full, we must keep these group tokens in full-precision. KIVI also maintain the local tokens in full-precision because these tokens are found important in many studies, we follow this operation and call these tokens recent tokens.

## 4 EXPERIMENTS

### 4.1 SETTINGS

**Baselines and Models.** In this paper, we focus on tuning-free KV Cache quantization methods. To our knowledge, KIVI is currently the strongest tuning-free baseline with the best compression efficiency and accuracy. Therefore, to assess the effectiveness of our method, we compare *SinkQ* with KIVI (Liu et al., 2024b) and vanilla FP16 implementation using greedy decoding across two famous model families: LLaMA (Touvron et al., 2023b; Dubey et al., 2024) and Mistral (Jiang et al., 2023). Specifically, we select LLaMA2-7B-chat-hf, LLaMA2-13B-chat-hf, LLaMA3-8B-Instruct, and Mistral-7B-Instruct-v0.2. Additional experiment results on other baselines and models can be found in Appendix B.

**Tasks.** We evaluate our methods on two benchmarks according to the length of input texts. For normal context length evaluation, we use arithmetic reasoning task Gsm8k (Cobbe et al., 2021), mainstream language and symbolic reasoning task BBH (Suzgun et al., 2023), and code completion task HumanEval (Chen et al., 2021) with different settings. For long context length evaluation, we choose four types of tasks in LongBench (Bai et al., 2024) including Document QA (Qasper), Sum-

Table 1: Results of the performance on GSM8K, BBH, and HumanEval (HE). **Bold** indicates the best results. We report accuracy for Gsm8k, BBH and Pass@k for HumanEval. Pass@k (p@k) refers to running each test question k times and calculating the average pass rate of the generated code. *SinkQ* outperforms KIVI across all tasks, achieving the best results.

| Dataset | LLaMA2-7B-chat-hf | | | LLaMA2-13B-chat-hf | | | LLaMA3-8B-Instruct | | | Mistral-7B-Instruct | | |
|---|---|---|---|---|---|---|---|---|---|---|---|---|
| | Fp16 | KIVI | Ours | Fp16 | KIVI | Ours | Fp16 | KIVI | Ours | Fp16 | KIVI | Ours |
| Gsm8k (8) | 21.99 | 16.30 | **21.38** | 36.54 | 28.51 | **36.09** | 74.91 | 63.15 | **72.55** | 42.91 | 37.38 | **41.17** |
| + CoT | 21.30 | 17.51 | **18.20** | 37.00 | 31.77 | **36.92** | 76.72 | 66.79 | **75.06** | 42.99 | 37.45 | **41.39** |
| + 0-CoT | 24.11 | 21.61 | **22.59** | 32.60 | 29.19 | **31.31** | 40.64 | 37.54 | **42.68** | 40.18 | 33.81 | **37.98** |
| BBH (3) | 33.34 | 32.48 | **33.36** | 37.61 | 36.20 | **37.43** | 45.77 | 44.19 | **45.60** | 42.10 | 40.29 | **42.02** |
| + CoT | 40.21 | 34.00 | **35.17** | 47.38 | 41.02 | **44.37** | 68.18 | 47.38 | **60.31** | 51.33 | 36.42 | **41.93** |
| + 0-CoT | 35.00 | 33.30 | **34.25** | 35.86 | 33.57 | **34.80** | 51.37 | 44.19 | **48.89** | 41.74 | 37.83 | **40.19** |
| HE (p@1) | 12.19 | 9.75 | **11.58** | 7.92 | 7.31 | **7.92** | 40.24 | 28.05 | **40.85** | 40 .24 | 32.92 | **35.36** |
| HE (p@10) | 17.07 | 12.19 | **14.63** | 13.41 | 11.58 | **15.24** | 69.51 | 56.09 | **67.68** | 54.87 | 50.00 | **54.26** |
| Average | 25.65 | 22.14 | **23.90** | 31.04 | 28.14 | **30.51** | 58.42 | 48.16 | **56.70** | 44.55 | 38.26 | **41.79** |

marization (GovReport, MultiNews), Few-shot Learning (TriviaQA, SamSum, TREC) and Code completion (LCC, RepoBench-P). We focus on downstream task performance rather than language modeling abilities such as perplexity (PPL).

**Details.** We implement both KIVI and *SinkQ* under 2-bit quantization. For KIVI, the group size (G) and residual length (R) are set to 128. For *SinkQ*, we use G = 128, R = 32, and set $sink\_num$ to 3. Notably, we set $sink\_num$ to 0 for the first and second layers because we find that shallow layers have no attention sinks (Ablation in Section 4.4). Regarding the sinks that have been eliminated from the sink pool, we do not put these tokens back in their original positions, but still keep a full precision window to store these sinks for easier implementation. Our experiments find that a very small window can retain all the eliminated sinks, and we set it to 32. GSM8K and BBH are tested under the LM Eval (Gao et al., 2024) framework. Humaneval follows the settings from InstructEval[2]. Additionally, we use a CUDA fused kernel from (Dettmers et al., 2022) for efficient multiplication of full-precision and quantized matrices in both KIVI and *SinkQ*. All the experiments are conducted on NVIDIA A100 40G GPUs unless otherwise specified.

## 4.2 RESULTS

### 4.2.1 NORMAL CONTEXT LENGTH EVALUATION

Table 1 presents the results of the normal context length evaluation across different models and methods. For Gsm8k and BBH, we report accuracy in the setting of few-shot, few-shot CoT, and zero-shot CoT. For HumanEval, we report pass@1 and pass@10 in the zero-shot setting. The results illustrate that our method significantly outperforms KIVI across all settings. Notably, on BBH (3-CoT, LLaMA3-8B-Instruct), *SinkQ* achieves a 12.93% improvement over KIVI. Compared to FP16, *SinkQ* incurs minor accuracy loss in most settings. The largest accuracy drop occurs on BBH (3-CoT, Mistral-7B-Instruct), likely due to the high complexity of the task and the long generation length required. Overall, *SinkQ* can achieve significant performance improvements over the best existing baseline.

### 4.2.2 LONG CONTEXT LENGTH TASKS EVALUATION

The main results of long context length evaluation are in table 2. Our method outperforms KIVI in most settings, with only a tiny performance gap compared to the FP16 baseline. While KIVI

---

[2]https://github.com/declare-lab/instruct-eval

Table 2: Main results on LongBench. We report accuracy for TREC, Rouge-L for GovReport and SamSum, edit similarity (Levenshtein distance (Svyatkovskiy et al., 2020)) for LCC and Re-poBenchP, and F1 score for the other tasks. **Bold** indicates the best results for each setting. *SinkQ* demonstrates superior performance on average and exhibits nearly lossless compression accuracy.

| Model | | Qasper | GovReport | MultiNews | TREC | TriviaQA | SamSum | LCC | RepoBench-P | Avg |
|---|---|---|---|---|---|---|---|---|---|---|
| | Fp16 | 20.04 | 25.08 | 23.02 | 59.67 | 85.39 | 39.28 | 59.59 | 48.04 | 45.01 |
| LLaMA2-7B-chat-hf | KIVI | **20.43** | 19.97 | 19.82 | 59.67 | **85.16** | 37.70 | 58.73 | 47.24 | 43.59 |
| | Ours | 19.95 | **21.56** | **20.81** | 59.67 | 85.00 | **39.10** | 59.44 | **48.51** | **44.26** |
| | Fp16 | 17.42 | 25.65 | 23.35 | 64.00 | 86.52 | 40.49 | 49.80 | 47.13 | 44.30 |
| LLaMA2-13B-chat-hf | KIVI | **20.10** | 20.65 | 21.10 | 63.67 | 86.39 | 39.51 | 49.10 | 43.95 | 43.06 |
| | Ours | 18.81 | **22.29** | **21.69** | 64.00 | **86.81** | 40.35 | 51.14 | **47.71** | **44.10** |
| | Fp16 | 37.54 | 31.04 | 25.58 | 69.67 | 89.85 | 40.50 | 56.58 | 51.01 | 50.22 |
| LLaMA3-8B-Instruct | KIVI | 34.88 | 28.43 | 24.78 | 69.33 | 89.57 | 40.09 | 44.42 | 45.54 | 47.13 |
| | Ours | **36.75** | **30.74** | **24.94** | 69.67 | 89.74 | 40.39 | 52.37 | 48.82 | **49.18** |
| | Fp16 | 24.35 | 33.05 | 25.77 | 67.00 | 86.84 | 40.95 | 57.24 | 49.84 | 48.13 |
| Mistral-7B-Instruct | KIVI | **24.20** | 30.98 | 25.10 | 66.33 | 85.40 | 41.05 | 55.70 | 48.18 | 47.12 |
| | Ours | 23.78 | **31.37** | **25.35** | 66.33 | 86.18 | 41.25 | 55.89 | 48.32 | **47.31** |

maintains good accuracy on most tasks, it occasionally experiences significant performance drops (e.g., LLaMA3-8B Instruct, LCC: 56.58% → 44.42%). However, *SinkQ* does not encounter this situation, which suggests that our method achieves higher quantization accuracy than KIVI.

## 4.3 EFFICIENCY COMPARISON

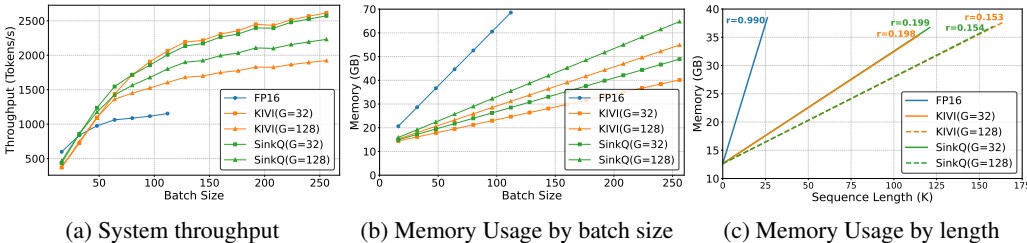

(a) System throughput     (b) Memory Usage by batch size     (c) Memory Usage by length

Figure 3: Experiments on throughput and memory: (a) Comparison of throughput (tokens/s) for different methods across different batch sizes on NVIDIA A800 80G. (b) Peak memory usage (including model weights and other components) at different batch sizes on NVIDIA A800 80G. (c) Peak memory usage (including model weights and other components) at different sequence lengths when batch size = 1 on NVIDIA A100 40G. *SinkQ* achieves a peak memory reduction of up to 6.4× and a throughput increase of 2.3×.

Additionally, to validate the memory reduction and throughput improvements achieved by *SinkQ*, we conduct three experiments: a throughput test, a memory test, and a longest sentence test. The throughput test measures the number of tokens generated per second as the batch size varies while keeping the input and output lengths fixed. The memory test tracks memory usage as the batch size changes, also with fixed input and output lengths. The longest sentence test assesses the memory required for inference as the output length increases infinitely (until out-of-memory), with a fixed batch size of 1 and an input length of 1. We use the LLaMA2-7B-chat-hf model for our experiments, and set the input length to 64 and the output length to 384 for both the throughput and memory tests. Figure 3 illustrates the results.

Figure 3a shows that when the batch size is small, *SinkQ* performs slightly slower than the FP16 baseline. However, as the batch size increases, *SinkQ* demonstrates a significant speed advantage. Compared to KIVI, *SinkQ* is slightly slower at $G = 32$ but significantly faster at $G = 128$. This

Table 3: Ablation study of *SinkQ* by combining possible group sizes G and residual lengths R. The settings in the main experiment are indicated with underlines.

| G | R | Gsm8k(8) | Gsm8k(8-CoT) | G | R | Gsm8k(8) | Gsm8k(8-CoT) | G | R | Gsm8k(8) | Gsm8k(8-CoT) |
|---|---|---|---|---|---|---|---|---|---|---|---|
| 32 | 0 | 70.05 | 73.16 | 64 | 0 | 68.92 | 72.63 | 128 | 0 | 70.96 | 73.54 |
| 32 | 8 | 71.95 | 74.22 | 64 | 8 | 70.05 | 73.01 | 128 | 8 | 72.51 | 74.15 |
| 32 | 16 | 72.78 | 74.83 | 64 | 16 | 70.89 | 73.84 | 128 | 16 | 72.93 | 74.37 |
| 32 | 32 | 72.78 | 74.53 | 64 | 32 | 72.40 | 75.06 | 128 | 32 | 72.55 | 75.06 |
| 32 | 64 | 74.00 | 76.88 | 64 | 64 | 73.69 | 76.42 | 128 | 64 | 73.24 | 75.36 |
| 32 | 128 | 73.77 | 77.33 | 64 | 128 | 74.68 | 76.42 | 128 | 128 | 73.24 | 76.65 |

variation can be attributed to the difference in their handling of recent tokens and factors such as quantization timing and sequence length. From Figure 3b, it is evident that quantization can significantly reduce memory usage compared to the FP16 baseline. While *SinkQ* requires slightly more memory than KIVI, this can also be attributed to differences in their management of recent tokens. Additionally, with larger group sizes, more full-precision tokens must be retained when the group is not full, leading to increased memory requirements. This phenomenon becomes more pronounced with large batch sizes and short sequence lengths. Figure 3c allows us to observe the compression ratio (the slope of each line) of the KV Cache more clearly. When the sequence length is sufficiently large, the influences of the group and recent tokens become negligible. Notably, when $G = 32$, the compression ratio is approximately 5.0 times; at $G = 128$, it reaches about 6.4 times.

### 4.4 ABLATION

**Group size and residual length.** Group size and residual length are critical hyperparameters in *SinkQ*. Theoretically, a larger group size allows more values to be quantized at each step, which can reduce quantization accuracy due to the increased range of $X_{max} - X_{min}$. On the other hand, a larger group size decreases memory usage by requiring fewer quantization coefficients to be retained. Conversely, increasing the residual length requires more memory since a more full-precision KV Cache must be retained, but it also improves accuracy. Thus, selecting an appropriate group size and residual length is critical to balancing memory usage and accuracy.

We explore the impact of group size and residual length with group sizes of {32, 64, 128} and residual lengths of {0, 8, 16, 32, 64, 128}. Table 3 reports the results for LLaMA3-8B-Instruct on Gsm8k 8-shot and 8-shot CoT under different configurations. When the group size is fixed, we observe a clear upward trend in accuracy as the residual length increases. However, when the residual length is fixed, the effect of group size shows no clear pattern, likely because the token distribution is relatively uniform, meaning that increasing group size has a limited impact. Since increasing the group size can improve the compression ratio (if not consider the group tokens), we tend to choose a larger group size. For our main experiments, we choose a group size of 128 and a residual length of 32 to balance performance and compression ratio.

**The number of sinks.** We explore the effect of varying the number of sinks ($sink\_num$) from 0 to 6, keeping all other settings unchanged. Table 4 presents the results for LLaMA3-8B-Instruct on Gsm8k (8-shot and 8-shot CoT). The results show that retaining even a single sink can significantly improve performance, but further increases in $sink\_num$ yield diminishing returns, eventually plateauing performance. However, the increase in $sink\_num$ may result in more memory overhead, leading to a decrease in compression ratio. Considering that a small $sink\_num$ is already sufficient to significantly improve the accuracy, we set $sink\_num = 3$ for our main experiments, which is also consistent with our previous explorations.

**Sinks in shallow layers.** We observe that there are no attention sinks in the shallow layers (see Figure 5 and Figure 6 in Appendix, the Keys shallow layers does not exhibit the characteristics discussed in Section 3.1.), suggesting that $sink\_num$ should be set to 0 in these layers. To explore this further, we set $sink\_num$ to 0 in consecutive shallow layers and evaluate the performance on Gsm8k (8-shot and 8-shot CoT) using LLaMA3-8B-Instruct. For example, "0 $\sim$ 2" means that $sink\_num$ is set to 0 for the first three layers of the model. Table 4 shows that the impact is minimal

Table 4: Ablation study of $sink\_num$. The settings in the main experiment are indicated with underlines. (left) Results on Gsm8k with different $sink\_num$. (right) Results on Gsm8k with $sink\_num = 0$ in shallow layers.

| $sink\_num$ | Gsm8k(8) | Gsm8k(8-CoT) | Layers | Gsm8k(8) | Gsm8k(8-CoT) |
|---|---|---|---|---|---|
| 0 | 62.09 | 68.31 | None | 72.48 | 75.59 |
| 1 | 71.80 | 75.74 | 0 | 72.78 | 75.44 |
| 2 | 71.57 | 75.06 | 0,1 | 72.55 | 75.06 |
| 3 | 72.55 | 75.06 | $0 \sim 2$ | 71.49 | 74.68 |
| 4 | 72.25 | 75.97 | $0 \sim 3$ | 71.80 | 73.64 |
| 5 | 72.18 | 75.74 | $0 \sim 4$ | 70.96 | 74.53 |
| 6 | 72.18 | 75.89 | $0 \sim 5$ | 69.60 | 74.00 |

in the shallowest layers but becomes more significant as we move deeper into the model. Based on these results, we set $sink\_num = 0$ for the first two layers in all models for our main experiments.

## 5 RELATED WORK

**Efficient Inference of LLMs.** Large Language Models often have enormous parameters, leading to significant computational costs during inference. To address this, some researchers have employed parameter pruning techniques to eliminate redundant or less important parameters, thereby compressing LLMs (Ma et al., 2023; Xia et al., 2024; Frantar & Alistarh, 2023). Other studies have focused on quantizing model weights, reducing their size and the number of arithmetic operations required for inference. For example, GPTQ (Frantar et al., 2022) uses second-order information to quantize models to 3 or 4-bit precision while maintaining accuracy. AWQ (Lin et al., 2024) preserves critical weights based on the activation distribution, quantizing the remaining weights to lower bit precision. These methods can be combined with KV Cache compression to achieve a better memory usage and a higher throughput.

**KV Cache Compression.** KV Cache compression can significantly reduce the size of KV Cache with minimal accuracy loss. Liu et al. (2024b) find that some outlier channels in the Keys have very large magnitudes, resulting in a significant loss when quantifying Keys by token. They further discover that Keys are distributed by channel, while Values are distributed by token. Based on these observations, they introduce KIVI, a tuning-free 2-bit quantization method that improve quantization accuracy by applying channel-wise quantization to the Key cache and token-wise quantization to the Value cache. Hooper et al. (2024) find that quantizing the Key cache before applying rotary positional embeddings reduces the negative impact of quantization. GEAR (Kang et al., 2024) compensates for compression-induced errors by combining low-rank and sparse matrices, achieving near-lossless results in 4-bit quantization, but it will bring additional calculations due to the low-rank matrix calculation. Additionally, evicting some tokens during inference helps reduce the excessive memory usage caused by storing KV Cache. Xiao et al. (2024) propose StreamingLLM, which retains the initial and final tokens of the input, leveraging the concept of the *attention sink*. Moreover, Zhang et al. (2023) consider that only a minority of tokens significantly influence the output and that these critical tokens are positively correlated with their frequency of occurrence. They propose H2O, which retains recent tokens while dynamically evicting less important ones.

## 6 CONCLUSION

In this paper, we start from the assumptions of KIVI and further explore the distribution of the Keys in the outlier channels. We observe that several unusual tokens deviate from the assumptions of KIVI and exhibit exceptionally high attention scores. Quantizing these tokens has detrimental effects, as it increases the quantization errors of other tokens, and their high attention scores lead to more significant quantization errors reflected in the attention output. Building on these observations, we propose Sink-Aware KV Cache quantization (*SinkQ*), which leverages the magnitude of the

Keys to dynamically track these tokens during decoding, excluding them from the quantization process while retaining their full-precision values. Extensive experiments show that our method achieves significant improvements in accuracy, along with substantial reductions in memory usage and increases in throughput.

# 7 LIMITATIONS & FUTURE WORK

Although *SinkQ* has achieved excellent results, there are still some limitations:

- Due to the presence of the group and recent tokens, we cannot ensure that all tokens are quantized at every moment. When the sequence length is very short and the batch size is very large, the compression ratio of *SinkQ* is reduced. In extreme cases, when the sequence length is shorter than the group size, *SinkQ* does not perform any compression.
- *SinkQ* occasionally still incurs a little loss on specific datasets under 2-bit quantization. This may be related to the difficulty of the datasets and the required generation length. When the generation length is very long, *SinkQ* may face an unacceptable risk of loss due to error accumulation.

Additionally, there are still some points worth exploring:

- Mitigating error accumulation: When the generation length is excessively long, KV Cache quantization may lead to unacceptable errors due to error accumulation. Developing techniques to alleviate this issue is a promising direction.
- Combining with other KV Cache compression techniques: KV Cache quantization is compatible with other compression methods. Combining it with other KV Cache compression techniques could achieve State-of-the-art results.

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

# A ADDITIONAL FIGURES AND TABLES.

| Input: |
|---|
| [INST]Which methods did Socrates employ to challenge the prevailing thoughts of his time?
[/INST] |

Output:

[INST]Which methods did Socrates employ to challenge the prevailing thoughts of his time?

[/INST] Socrates employed several methods to challenge the prevailing thoughts of his time, including:

1. Questioning: Socrates used questioning as a primary method to challenge the beliefs of others. He would ask probing questions to expose the contradictions and inconsistencies in their arguments, and to lead them to a deeper understanding of their own beliefs.

2. Dialogue: Socrates believed in the power of dialogue to promote critical thinking and understanding. He would engage in conversations with others, using his questions and their responses to challenge their beliefs and to encourage them to think more deeply about their ideas.

3. Irony: Socrates often used irony to challenge the beliefs of others. By using irony, he could expose the flaws in their arguments and lead them to see things from a different perspective.

4. Paradox: Socrates often used paradoxical statements to challenge the prevailing beliefs of his time. By presenting seemingly contradictory statements, he could encourage his interlocutors to think more critically about their beliefs and to question their assumptions.

5. Maieutics: Socrates believed in the power of midwifery, or maieutics, to bring forth new ideas and insights. He would use his questions and dialogue to help others discover new truths and to challenge their existing beliefs.

6. Analogy: Socrates often used analogies to help others understand complex ideas and to challenge their beliefs. By using comparisons and contrasts, he could help his interlocutors to see things from a different perspective and to develop a deeper understanding of the issues at hand.

7. Refutation: Socrates was skilled at refuting the arguments of others by exposing their flaws and inconsistencies. He would use his questions and dialogue to challenge the beliefs of others and to lead them to a deeper understanding of the issues at hand.

8. Affirmation: Socrates also used affirmation to challenge the prevailing thoughts of his time. By affirming the beliefs of others and building on them, he could encourage them to think more deeply about their ideas and to develop a more nuanced understanding of the issues at hand.

9. Silence: Socrates believed that sometimes the best way to challenge the prevailing thoughts of his time was through silence. By remaining silent in the face of foolish or false arguments, he could encourage others to think more critically about their beliefs and to question their assumptions.

Overall, Socrates employed a wide range of methods to challenge the prevailing thoughts of his time, including questioning, dialogue, irony, paradox, maieutics, analogy, refutation, affirmation, and silence. Through these methods, he was able to encourage critical thinking and to promote a deeper understanding of the issues at hand.

Table 5: Example generated by LLaMA-2-7B-chat-hf.

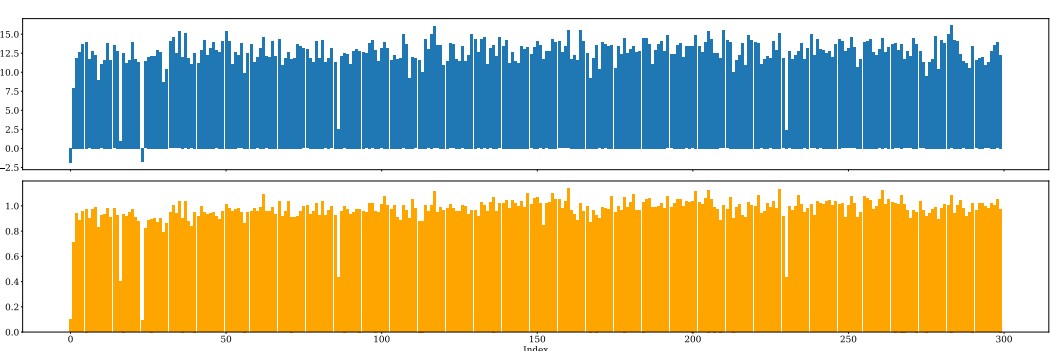

Figure 4: The Keys in an outlier channel (up) and the magnitude of the Keys overall (down).

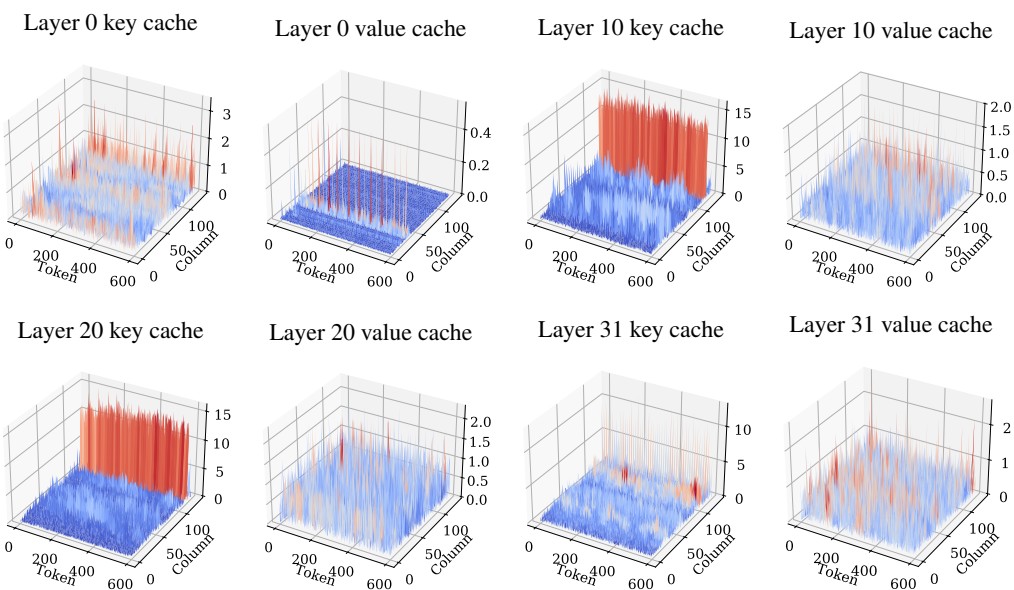

Figure 5: Magnitude of the keys and Values for Llama-2-7B-chat-hf in head 17.

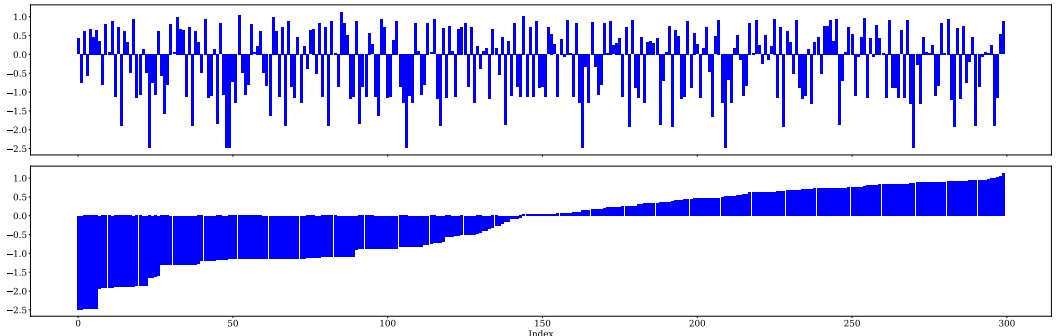

Figure 6: The Keys in an outlier channel (up) and the sorted Keys in an outlier channel (down).

# B ADDITIONAL EXPERIMENT RESULTS.

## B.1 EXPERIMENTS ON LLAMA-2-70B-CHAT-HF.

| 70b-chat-hf | Gsm8k(8) | Gsm8k(8-cot) | Gsm8k(0-cot) | BBH(3) | HE(p@1) | Avg |
|---|---|---|---|---|---|---|
| FP16 | 56.03 | 55.04 | 48.98 | 47.09 | 16.46 | 44.72 |
| KIVI | 51.63 | 50.49 | 46.40 | 46.08 | 14.02 | 41.72 |
| Ours | **52.92** | **52.54** | **49.05** | **46.48** | **15.85** | **43.37** |

Table 6: Experiments on LLaMA-2-70b-chat-hf

To validate the performance on larger models, we conducte some additional experiments on LLaMA-2-70b-chat-hf. Due to time and computational limitations, we only add experimental results from a portion of the dataset on our main experiments. The experimental setup is completely consistent with the main experiment. The result shows that SinkQ can still achieve higher accuracy advantages on larger models based on KIVI.

## B.2 COMPARISON WITH TOKEN EVICTION METHODS.

| Llama2-7b-chat | Qasper | GovReport | MultiNews | TREC | TriviaQA | SamSum | LCC | Repobench-P | Avg |
|---|---|---|---|---|---|---|---|---|---|
| FP16 | 20.04 | 25.08 | 23.02 | 59.67 | 85.39 | 39.28 | 59.59 | 48.04 | 45.01 |
| KIVI | **20.43** | 19.97 | 19.82 | 59.67 | **85.16** | 37.7 | 58.73 | 47.24 | 43.59 |
| SnapKV | 18.96 | 18.73 | 19.64 | 59 | 84.84 | 38.22 | **60.5** | **50.08** | 43.75 |
| H2O | 17.51 | 18.85 | 19.88 | 50 | 84.22 | 38.09 | 58.23 | 49.66 | 42.05 |
| Streaming | 15.31 | 19.39 | 18.99 | 51 | 83.11 | 36.8 | 57.57 | 47.33 | 41.19 |
| Ours | 19.95 | **21.56** | **20.81** | **59.67** | 85 | **39.1** | 59.44 | 48.51 | **44.26** |

Table 7: Experiments on three additional eviction-based methods on LLaMA-2-7b-chat-hf.

We add some comparisons with the token eviction methods. The previous token eviction methods are mostly evaluated on LongBench, so we also conduct experiments on LongBench using LLaMA-2-7b-chat-hf. The input length of LongBench is relatively long, while the output length is relatively short, which may be more conducive to the performance of the token eviction methods. The baselines include StreamingLLM, H2O, and SnapKV. In order to maintain the simplicity and consistency of the settings for comparison, we only perform token eviction in the prefill stage, and retain all KV caches in the decode stage. In addition, we make some adjustments to H2O based on SnapKV's strategy, selecting only the queries in the sliding window for attention score selection (which was later verified to be superior to H2O's strategy). In order to maintain the overall compression ratio consistent with SinkQ, we choose to evict 84% of the tokens in the prefill stage, which have the closest compression ratio to SinkQ. For H2O, the number of recent tokens and heavy hitters is the same. For StreamingLLM, we do not adjust its position id during decoding phase. So, the process of token eviction is as follows:

- In the prefill stage, use queries in the sliding window to calculate the attention score with other tokens, and perform token eviction according to the strategies of StreamingLLM, H2O, and SnapKV respectively.

- During the decode phase, attention calculation is performed directly without token eviction.

Among these methods, SnapKV achieves the best results. But even under more favorable settings, the result is still slightly lower than SinkQ.

## B.3 COMPARISON WITH ADDITIONAL BASELINES.

| Llama2-7b-chat | Gsm8k(8) | 8-cot | 0-cot | BBH(3) | 3-cot | 0-cot | HE(p@1) | p@10 | Avg |
|---|---|---|---|---|---|---|---|---|---|
| FP16 | 21.99 | 21.3 | 24.11 | 33.34 | 40.21 | 35.00 | 12.19 | 17.07 | 45.01 |
| KIVI | 16.3 | 17.51 | 21.61 | 32.48 | 34 | 33.30 | 9.75 | 12.19 | 43.59 |
| Ours | **19.86** | **19.33** | **22.52** | **33.33** | **34.43** | **33.74** | **11.58** | **14.63** | **43.75** |
| ZipCache | 15.92 | 17.74 | 20.02 | 32.85 | 33.9 | 32.35 | 9.45 | 15.24 | 42.05 |

Table 8: Experiments on other baselines.

We add ZipCache as our baseline, and in order to maintain consistent compression rates, we set 20% of the tokens to 4-bit quantization and 80% to 2-bit quantization. The other hyper-parameters in ZipCache are the same. We conduct our experiments on GSM8K, BBH and HumanEval with LLaMA-2-7b-chat-hf. The results show that ZipCache is weaker than KIVI and SinkQ.

## C ADDITIONAL TIME ANALYSIS.

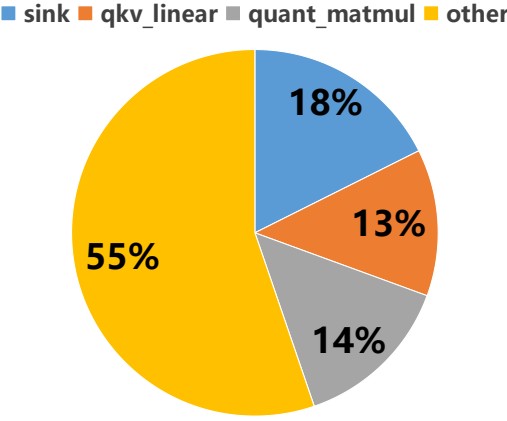

Figure 7: The time proportion of each part in the attention block.

We provide a more detailed analysis of the time cost. Firstly, we analyze the compression stage. In the compression stage, we calculate the magnitude of each token's key, perform threshold comparison, select the index, and quantify it. In the compress phase, the cost of the sink operation is relatively high compared to quantization, but the compress operation is only performed every G steps (128 in the main experiment), so this time cost can be almost negligible. In the attention calculation stage, we need to calculate the query, key and value in the sink pool and cover the attention score according to the sink index, which has a certain cost. We have drawn the main time overhead in Figure 9, and the sink operation accounts for about 18% of the time in the attention block. However, considering the pre-processing, post-processing and FFN calculation in the entire forward step, the time proportion of sink operations is very small overall.

# D ACCUMULATIVE ATTENTION SCORE.

In order to more intuitively display the proportion of attention scores of attention sinks, we sort the accumulative attention scores and plot them on a graph. Due to the significant difference, we also plot the results after log2 mapping.

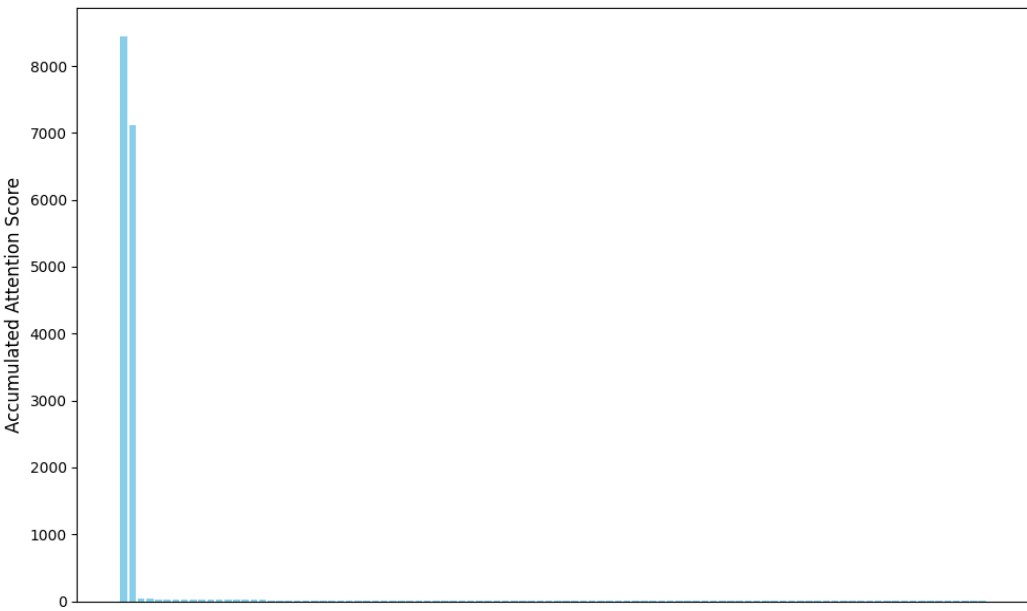

Figure 8: Accumulated attention score.

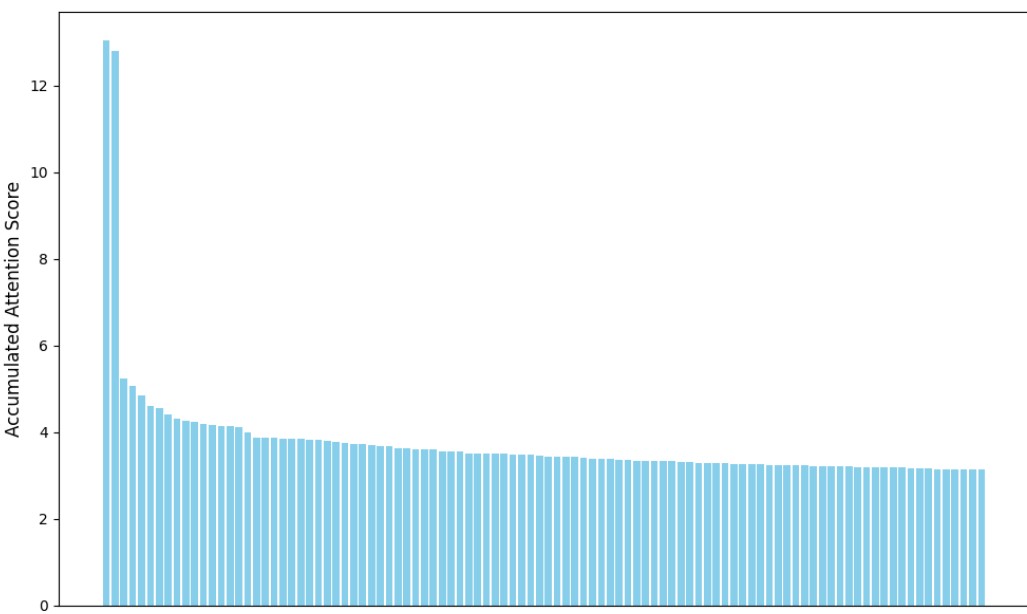

Figure 9: Accumulated attention score (log2).

