# OpenReview forum: "SinkQ: Accurate 2-bit KV Cache Quantization with Dynamic Sink Tracking"
_ICLR.cc/2025/Conference — ICLR 2025 Conference Withdrawn Submission_

### Official Review · Reviewer_sFw2 · 2024-10-18

**Soundness:** 3
**Presentation:** 3
**Contribution:** 3
**Rating:** 6
**Confidence:** 4

**Summary:**

This paper proposes SinkQ, a near 2-bit KV Cache quantization method that demonstrates quality improvement over previous SOTA while improving throughput and accuracy.

**Strengths:**

The target problem is clear and have both theoretical and empirical comparison with previous work.
The paper also conducted a detailed analysis on the attention values as well as controlled experiment to explain how some operations could increase accuracy.
It is a nice extension from previous method (KIVI) and handles some outlier token problem in that KIVI does not solve.

**Weaknesses:**

Writing: Authors did not explain enough of the baseline method of KIVI, especially the part of group and residual in KIVI. The method is partly based on KIVI but authors mainly explained how they improved upon KIVI. Adding background on how KIVI separates between grouping and residual will be nice.
Eval: Lacks comparison with other non-quantization works
Lacks throughput number for ablation studies
(minor) Lacks experiment to larger models (ex. 70B)

**Questions:**

Can you add some comparison with non-quantization based compression work? In the current paper the only baseline is KIVI and original.

---

> ### Author Response · Authors · 2024-11-27
>
> Thank you for your valuable comments. We will address your concerns as follows.
>
> Q:
>
> Writing: Authors did not explain enough of the baseline method of KIVI, especially the part of group and residual in KIVI. The method is partly based on KIVI but authors mainly explained how they improved upon KIVI. Adding background on how KIVI separates between grouping and residual will be nice.
>
> A:
>
> We have changed the related context in Section 3.2.
>
>
>
> Q:
>
>  Lacks comparison with other non-quantization works .
>
> A:
>
> Thanks for your suggestion. We have added the results of some non-quantization methods and some other quantization-based methods (ZipCache, H2O, Streamingllm and SnapKV). The results are shown below and the experimental details can be found in Appendix B.
>
> | Llama2-7b-chat | Gsm8k(8)  | 8-cot     | 0-cot     | BBH(3)    | 3-cot     | 0-cot     | HE(p@1)   | p@10      | Avg       |
> | -------------- | --------- | --------- | --------- | --------- | --------- | --------- | --------- | --------- | --------- |
> | FP16           | 21.99     | 21.3      | 24.11     | 33.34     | 40.21     | 35.00     | 12.19     | 17.07     | 45.01     |
> | KIVI           | 16.3      | 17.51     | 21.61     | 32.48     | 34.00     | 33.30     | 9.75      | 12.19     | 43.59     |
> | Ours           | **19.86** | **19.33** | **22.52** | **33.33** | **34.43** | **33.74** | **11.58** | **14.63** | **43.75** |
> | ZipCache       | 15.92     | 17.74     | 20.02     | 32.85     | 33.90     | 32.35     | 9.45      | 15.24     | 42.05     |
>
>
>
> | Llama2-7b-chat | Qasper    | GovReport | MultiNews | TREC      | TriviaQA  | SamSum   | LCC      | Repobench-P | Avg       |
> | -------------- | --------- | --------- | --------- | --------- | --------- | -------- | -------- | ----------- | --------- |
> | FP16           | 20.04     | 25.08     | 23.02     | 59.67     | 85.39     | 39.28    | 59.59    | 48.04       | 45.01     |
> | KIVI           | **20.43** | 19.97     | 19.82     | 59.67     | **85.16** | 37.7     | 58.73    | 47.24       | 43.59     |
> | SnapKV         | 18.96     | 18.73     | 19.64     | 59        | 84.84     | 38.22    | **60.5** | **50.08**   | 43.75     |
> | H2O            | 17.51     | 18.85     | 19.88     | 50        | 84.22     | 38.09    | 58.23    | 49.66       | 42.05     |
> | Streaming      | 15.31     | 19.39     | 18.99     | 51        | 83.11     | 36.8     | 57.57    | 47.33       | 41.19     |
> | Ours           | 19.95     | **21.56** | **20.81** | **59.67** | 85        | **39.1** | 59.44    | 48.51       | **44.26** |
>
> We are also reproducing the results of KVQuant, but there are some issues with the results. We hope we can attach the results of KVQuant before the deadline.
>
>
>
> Q:
>
> Lacks experiment to larger models (ex. 70B)
>
> A:
>
> For the scenario of larger models, we have added experiments of LLaMA-2-70b-chat-hf, the results are shown below and the experiment details can be found in Appendix B.
>
> | 70b-chat-hf | Gsm8k(8)  | Gsm8k(8-cot) | Gsm8k(0-cot) | BBH(3)    | HE(p@1)   | Avg       |
> | ----------- | --------- | ------------ | ------------ | --------- | --------- | --------- |
> | FP16        | 56.03     | 55.04        | 48.98        | 47.09     | 16.46     | 44.72     |
> | KIVI        | 51.63     | 50.49        | 46.40        | 46.08     | 14.02     | 41.72     |
> | Ours        | **52.92** | **52.54**    | **49.05**    | **46.48** | **15.85** | **43.37** |

---

> > ### Comment · Reviewer_sFw2 · 2024-11-27
> >
> > Thanks for the clarification. I think your work is above the bar although some explanation/comparison is not complete. Anyways this is good work and you should keep polishing it no matter whether it gets accepted.

---

### Official Review · Reviewer_GoAq · 2024-11-03

**Soundness:** 2
**Presentation:** 3
**Contribution:** 2
**Rating:** 5
**Confidence:** 4

**Summary:**

This paper presents SinkQ, a method of KV cache quantization that takes care of attention sinks, i.e., outlier tokens that exhibit significantly high attention scores. Different from methods that rely on uniform quantization or fixed attention sinks, SinkQ can dynamically track and store attention sinks that appear at any positions within a sentence. Evaluation results on normal context-length tasks and long-context tasks demonstrate that SinkQ can achieve better accuracy and a comparable level of throughput / memory usage as compared to KIVI (SoTA 2-bit quantization baseline).

**Strengths:**

`+` The design of the method mainly stems from insightful empirical observations (clearly presented and visualized). Thus, the motivation of the work is very clear. The design itself is also natural and many intuitions are aligned with observations from prior work.

`+` Eval results on normal context-length tasks are good (in terms of accuracy), indicating that the design of dynamic attention sinks is effective.

**Weaknesses:**

`-` The take-away from section 4.3 is unclear: When the choice of group size (G) / residual length (R) varies, would the throughput increase as compared to KIVI be consistent? The memory overhead also appears a bit concerning to me --- Would the additional memory overhead of SinkQ scale even more when sink_num goes up? The broader concern here is about scalability --- Please refer to "questions".

`-` Minor issues: Please consider fixing the y-axis of figures 1b and 1e, and the citation on line 210.

**Questions:**

Thank you for submitting this paper for ICLR. I have one concern below and I greatly appreciate it if the authors can offer some insights or clarification. I don't think it would require new experiment results:

- Scalability: I understand that it would be difficult to run experiments on larger LLMs (like the 70B+ models), but one missing point in the paper is how SinkQ would succeed (or fail) when the number of model parameters or the context length continuously go up. One potential consequence I can think of could be that the number of outlier tokens (i.e. attention sinks) grow quickly as the scale of the model / the context increase. In this case, would a small size of sink_num work? From table 4, it seems to me that using sink_num=1 should be sufficient, but that seems a bit too good to be true in most cases. If not, to support a larger size of sink_num, how would the extra memory overhead look like? --- This is one concern I have when I take a closer look at figure 3b. And if the memory overhead is too large and you have to use a small, fixed number of sinks (which means that you cannot store all outlier tokens), would SinkQ still be superior to KIVI in this case?

**Details Of Ethics Concerns:**

This paper does not raise any ethics concerns.

---

> ### Author Response · Authors · 2024-11-27
>
> Thank you very much for your feedback. We believe that you have two main concerns:
>
> 1. How do throughput and memory change when hyperparameters change?
> 2. Can SinkQ maintain its performance in other scenarios, such as long sequences and larger models with a small sink num?
>
>
>
> Q:
>
> How do throughput and memory change when hyper-parameters change?
>
> A:
>
> We will provide a more detailed explanation of the memory usage during inference.
> Firstly, KIVI will have the following full precision tokens:
>
> 1. When the group is not full, the Keys in the group will be retained.
> 2. Recent Values.
>
> SinkQ will have the following full precision tokens:
>
> 1. When the group is not full, the Keys and Values in the group will be retained
> 2. Recent Keys and Values.
> 3. A very small number of sinks
>
> So, the main difference is that there are some differences between groups and recent in full precision tokens. We choose this processing method mainly to preserve the quantization consistency of k and v for ease of engineering implementation. And there seem to be some differences of  KIVI in the paper and code so we choose to fulfill it by ourself. To achieve fair comparison, we use G=128, R=32 for SinkQ and G=128, R=128 for KIVI in the main experiments.
>
>
>
> So, come back to your question:
>
> 1. When the choice of group size (G) / residual length (R) varies, would the throughput increase as compared to KIVI be consistent?
>
> The answer is yes.
>
> 2. Would the additional memory overhead of SinkQ scale even more when sink_num goes up?
>
> The answer is yes.
>
>
>
>
>
> Q:
>
> Can SinkQ maintain its performance in other scenarios, such as long sequences and larger models?
>
> A:
>
> For the scenario of long sequences, we give a brief description to justify that a small sink num is enough. The sum of attention scores is always 1 regardless of the length of the sequence. Tokens with extreme high attention scores in a long sequence cannot be more than that in a short sequence from a probability perspective .
>
> For the scenario of larger models, we have added experiments of LLaMA-2-70b-chat-hf, the results are shown below and the experiment details can be found in Appendix B. We have also included comparisons with other baselines in Appendix B.
>
> | 70b-chat-hf | Gsm8k(8)  | Gsm8k(8-cot) | Gsm8k(0-cot) | BBH(3)    | HE(p@1)   | Avg       |
> | ----------- | --------- | ------------ | ------------ | --------- | --------- | --------- |
> | FP16        | 56.03     | 55.04        | 48.98        | 47.09     | 16.46     | 44.72     |
> | KIVI        | 51.63     | 50.49        | 46.40        | 46.08     | 14.02     | 41.72     |
> | Ours        | **52.92** | **52.54**    | **49.05**    | **46.48** | **15.85** | **43.37** |
>
>
>
> So, come back to your concrens:
>
> 1. How SinkQ would succeed (or fail) when the number of model parameters or the context length continuously go up?
>
> Answer: SinkQ can succeed in these scenarios.
>
> 2. If the memory overhead is too large and you have to use a small, fixed number of sinks (which means that you cannot store all outlier tokens), would SinkQ still be superior to KIVI in this case?I hope these can address your concerns.
>
> Answer: SinkQ is definitely superior to KIVI because it preserves more full-precision tokens than KIVI.

---

### Official Review · Reviewer_MqTg · 2024-11-03

**Soundness:** 2
**Presentation:** 2
**Contribution:** 1
**Rating:** 3
**Confidence:** 4

**Summary:**

The paper presents SINKQ, a KV Cache quantization designed for efficient deployment of large language models (LLMs) by balancing memory use and accuracy. SINKQ identifies unusual tokens with distinct distribution and higher attention scores, which often lead to quantization errors.

**Strengths:**

The paper introduces an approach to improve quantization accuracy by dynamically excluding high-error tokens.

**Weaknesses:**

1. The major claim of this paper lacks sufficient justification. The foundation of the work is based on the statement, "Previous work suggests that attention sinks occur only in the initial tokens; however, our observations reveal that they can appear at any position within a sentence." However, there is no concrete evidence or examples provided to support this strong claim. To strengthen the paper, it would be beneficial if the authors included specific evidence, such as visualizations or quantitative analysis, to demonstrate instances where attention sinks appear at various positions within a sentence.
2. In the experiments, Figure 3(b) and Figure 3(c) suggest that the proposed method may not be as memory-efficient as Kiwi, despite claims of improved efficiency. It would be helpful for the authors to directly address this observation and clarify the reasons behind this apparent discrepancy. A more detailed analysis of the trade-offs between the proposed method and Kiwi in terms of memory efficiency would strengthen the discussion.
3. There are other state-of-the-art KV cache methods, such as those referenced in [1] and [2], which should be considered as additional baselines. Including these baselines would provide a more comprehensive evaluation. It would also be beneficial for the authors to explain why these specific baselines were not included initially and how their method compares to the key innovations introduced in [1] and [2].

[1] Hooper C, Kim S, Mohammadzadeh H, et al. Kvquant: Towards 10 million context length llm inference with kv cache quantization[J]. arXiv preprint arXiv:2401.18079, 2024.
[2] Zhang T, Yi J, Xu Z, et al. KV Cache is 1 Bit Per Channel: Efficient Large Language Model Inference with Coupled Quantization[J]. arXiv preprint arXiv:2405.03917, 2024.

**Questions:**

1. Will the code be made publicly available for reproducibility?
2. Can SINKQ be extended for higher-bit quantization, or is its design specifically optimized for low-bit scenarios?

---

> ### Author Response · Authors · 2024-11-27
>
> Thank you for your valuable comments. We will address your concerns as follows.
>
>
>
> Q:
>
> The major claim of this paper lacks sufficient justification. The foundation of the work is based on the statement, "Previous work suggests that attention sinks occur only in the initial tokens; however, our observations reveal that they can appear at any position within a sentence." However, there is no concrete evidence or examples provided to support this strong claim. To strengthen the paper, it would be beneficial if the authors included specific evidence, such as visualizations or quantitative analysis, to demonstrate instances where attention sinks appear at various positions within a sentence.
>
>
>
> A:
>
> We have provided an example of key visualization in Figure 1 (b). Additionally, we have randomly selected 10 examples in GSM8K and provided the positions of the four tokens with the highest attention scores:
>
> | id   | 0    | 1    | 2    | 3    | 4    | 5    | 6    | 7    | 8    | 9    |
> | ---- | ---- | ---- | ---- | ---- | ---- | ---- | ---- | ---- | ---- | ---- |
> | top1 | 0    | 0    | 0    | 0    | 0    | 0    | 0    | 0    | 0    | 0    |
> | top2 | 14   | 20   | 12   | 17   | 9    | 13   | 15   | 11   | 36   | 16   |
> | top3 | 120  | 81   | 47   | 36   | 8    | 36   | 55   | 83   | 16   | 15   |
> | top4 | 29   | 93   | 70   | 51   | 33   | 48   | 11   | 34   | 14   | 5    |
>
> Generally , [bos] token is attention sink, while attention sinks do not have obvious distribution patterns, but it is true that initial tokens are more likely to obtain higher attention scores.
>
>
>
> Q:
>
> In the experiments, Figure 3(b) and Figure 3(c) suggest that the proposed method may not be as memory-efficient as Kiwi, despite claims of improved efficiency. It would be helpful for the authors to directly address this observation and clarify the reasons behind this apparent discrepancy. A more detailed analysis of the trade-offs between the proposed method and Kiwi in terms of memory efficiency would strengthen the discussion.
>
> A:
>
> We will provide a more detailed explanation of the memory usage during inference.
> Firstly, KIVI will have the following full precision tokens:
>
> 1. When the group is not full, the Keys in the group will be retained.
> 2. Recent Values.
>
> SinkQ will have the following full precision tokens:
>
> 1. When the group is not full, the Keys and Values in the group will be retained
> 2. Recent Keys and Values.
> 3. A very small number of sinks
>
> So, the main difference is that there are some differences between groups and recent in full precision tokens. We choose this processing method mainly to preserve the quantization consistency of k and v for ease of engineering implementation. And there seem to be some differences of  KIVI in the paper and code so we choose to fulfill it by ourself. To achieve fair comparison, we use G=128, R=32 for SinkQ and G=128, R=128 for KIVI in the main experiments.
> Due to the above reasons, there may be some differences between SinkQ and KIVI when the batch size is large. However, when the batch size is 1 and the seq length is long, the number of these full precision tokens is very small compared to the quantized tokens, and can be almost ignored. At this time, the memory of the two methods is relatively close.

---

> > ### Author Response · Authors · 2024-11-27
> > **part2**
> >
> > Q:
> >
> > There are other state-of-the-art KV cache methods, such as those referenced in [1] and [2], which should be considered as additional baselines. Including these baselines would provide a more comprehensive evaluation. It would also be beneficial for the authors to explain why these specific baselines were not included initially and how their method compares to the key innovations introduced in [1] and [2].
> >
> > A:
> >
> > Thanks for your suggestion. We did not select these baselines mainly because there were some differences in settings which could lead to unfair comparisons. We have added the results of ZipCache, H2O, Streamingllm and SnapKV. The results are shown below and the experimental details can be found in Appendix B.
> >
> > | Llama2-7b-chat | Gsm8k(8)  | 8-cot     | 0-cot     | BBH(3)    | 3-cot     | 0-cot     | HE(p@1)   | p@10      | Avg       |
> > | -------------- | --------- | --------- | --------- | --------- | --------- | --------- | --------- | --------- | --------- |
> > | FP16           | 21.99     | 21.3      | 24.11     | 33.34     | 40.21     | 35.00     | 12.19     | 17.07     | 45.01     |
> > | KIVI           | 16.3      | 17.51     | 21.61     | 32.48     | 34.00     | 33.30     | 9.75      | 12.19     | 43.59     |
> > | Ours           | **19.86** | **19.33** | **22.52** | **33.33** | **34.43** | **33.74** | **11.58** | **14.63** | **43.75** |
> > | ZipCache       | 15.92     | 17.74     | 20.02     | 32.85     | 33.90     | 32.35     | 9.45      | 15.24     | 42.05     |
> >
> >
> >
> > | Llama2-7b-chat | Qasper    | GovReport | MultiNews | TREC      | TriviaQA  | SamSum   | LCC      | Repobench-P | Avg       |
> > | -------------- | --------- | --------- | --------- | --------- | --------- | -------- | -------- | ----------- | --------- |
> > | FP16           | 20.04     | 25.08     | 23.02     | 59.67     | 85.39     | 39.28    | 59.59    | 48.04       | 45.01     |
> > | KIVI           | **20.43** | 19.97     | 19.82     | 59.67     | **85.16** | 37.7     | 58.73    | 47.24       | 43.59     |
> > | SnapKV         | 18.96     | 18.73     | 19.64     | 59        | 84.84     | 38.22    | **60.5** | **50.08**   | 43.75     |
> > | H2O            | 17.51     | 18.85     | 19.88     | 50        | 84.22     | 38.09    | 58.23    | 49.66       | 42.05     |
> > | Streaming      | 15.31     | 19.39     | 18.99     | 51        | 83.11     | 36.8     | 57.57    | 47.33       | 41.19     |
> > | Ours           | 19.95     | **21.56** | **20.81** | **59.67** | 85        | **39.1** | 59.44    | 48.51       | **44.26** |
> >
> > We are also reproducing the results of KVQuant, but there are some issues with the results. We hope we can attach the results of KVQuant before the deadline.
> >
> >
> >
> >
> >
> > Q:
> >
> > Will the code be made publicly available for reproducibility?
> >
> > A:
> >
> > Yes, we have prepared the code and will release it after anonymous peroid.
> >
> >
> >
> > Q:
> >
> > Can SINKQ be extended for higher-bit quantization, or is its design specifically optimized for low-bit scenarios?
> >
> > A:
> >
> > Yes, the method that performs well at low precision will definitely have better performance at high precision.

---

### Official Review · Reviewer_N3cZ · 2024-11-03

**Soundness:** 2
**Presentation:** 2
**Contribution:** 1
**Rating:** 3
**Confidence:** 4

**Summary:**

This paper proposes SinkQ, a KV cache quantization method for reducing memory overhead and improving efficiency of LLM inference. The authors observe the existence of sink tokens, which are a small subset of tokens that exhibit outlier characteristics and can occur at any token position, which significantly impact model accuracy. Based on this observation, the authors propose to keep a fixed-size pool for storing these sink tokens in full precision, while quantizing all other tokens following the KIVI approach. Empirical evaluations show that SinkQ mostly outperforms KIVI in preserving model quality.

**Strengths:**

1. This paper studies an important problem.
2. The proposed approach is well-motivated by the authors' observation.

**Weaknesses:**

1. The authors' observation of sink tokens is not new or novel. Previous works have observed the existence of heavy-hitter tokens [1] or salient tokens [2] in LLMs, which may occur at any token position. Hence, the claim on line 211 is inaccurate: "Previous work suggesting that attention sinks occur only in the initial tokens..."
2. The proposed method lacks novelty. The proposed mixed-precision approach for KV cache quantization is similar to previous works [2,3], which also identify outlier tokens in the KV cache and preserve in higher precision or full precision.
3. The "Method" section is incomplete, missing some important details. It is not clear how SinkQ identifies the sink tokens, which is a critical part of the proposed method. During inference, the tokens are ingested in a streaming manner, while the authors only describe how the outlier tokens can be identified after saving all keys in an offline manner on line 152.
4. Baselines are missing from the experiments. Since the authors adopt a mixed-precision quantization approach, mixed-precision baselines, such as KVQuant and ZipCache, are more relevant than the KIVI baseline. Moreover, the comparison with KIVI feels like an unfair comparison; since SinkQ directly adopts the KIVI approach (per-channel for keys and per-token for values, with residual cache) and adds sink tokens, the model quality will certainly outperform KIVI, but with memory and latency overheads. These overheads are not made very clear in Table 1 and 2.

References

[1] Zhang, Zhenyu, et al. "H2o: Heavy-hitter oracle for efficient generative inference of large language models." NeurIPS 2023.

[2] He, Yefei, et al. "ZipCache: Accurate and Efficient KV Cache Quantization with Salient Token Identification." NeurIPS 2024.

[3] Dong, Shichen, et al. "QAQ: Quality Adaptive Quantization for LLM KV Cache." arXiv preprint arXiv:2403.04643 (2024).

**Questions:**

Can the authors please clarify the concerns raised in Weaknesses?

---

> ### Author Response · Authors · 2024-11-27
>
> Thank you for your valuable comments. We will address your concerns as follows.
>
> Q:
>
> The authors' observation of sink tokens is not new or novel. Previous works have observed the existence of heavy-hitter tokens [1] or salient tokens [2] in LLMs, which may occur at any token position. Hence, the claim on line 211 is inaccurate: "Previous work suggesting that attention sinks occur only in the initial tokens..."
>
>
>
> A:
>
> For sink and heavy hitters, we provide further explanation:
> We believe that attention sinks and heavy hitters are not completely identical concepts, and their differences are as follows:
>
> 1. Attention sinks refer to tokens with consistently high attention scores during the decoding process, with only a few tokens (usually less than 4).
> 2. Heavy hitters are relatively important tokens in the decoding process compared to other tokens, typically accounting for over 10%-20% of all tokens.
> 3. Heavy hitters contain attention sinks, but there are differences in the distribution within tokens and the number of tokens. We have plotted the accumulative attention scores (sorted) in Appendix D, indicating that attention sinks are more unique than heavy hitters.
>
> Moreover, SinkQ reveals the outlier tokens in channel-wise key quantization, which was not mentioned in previous works. This is almost the most important observation in our work, and it plays a crucial role in the effectiveness of SinkQ (see Figure 1 (f)).
>
>
>
> Q:
>
> The proposed method lacks novelty. The proposed mixed-precision approach for KV cache quantization is similar to previous works [2,3], which also identify outlier tokens in the KV cache and preserve in higher precision or full precision.
>
>
>
> A:
>
> SinkQ can be seen as a mixed-precision quantization method, but our innovation lies in other aspects:
>
> 1. SinkQ reveals the outlier token phenomenon in channel-wise key quantization, which was not mentioned in previous work
> 2. The identification method of the outlier token in SinkQ is different from other methods. It directly uses the magnitude of the key as the evaluation metric, which does not require the calculation and storage of the accumulated attention score at each step, but only needs to be calculated at the compress step. This greatly speeds up efficiency, and this method is perfectly adapted to our preliminary exploration.
> 3. SinkQ only needs to retain a few outlier tokens to significantly increase quantization accuracy, while methods such as ZipCache require retaining more tokens which it considers important.
>
> We further explain why our recognition strategy, namely the magnitude of the key, is better than using accumulative attention score:
>
> 1. Selecting outlier tokens based on the magnitude of the key can accurately identify them in the channel-wise key quantization, while the accumulative attention score may not (the top k of the attention score in Figure 1 (c) does not completely overlap with the top k of the magnitude of the key).
> 2. The calculation of accumulative attention score may not be accurate under this setting. Calculating based on the global attention score may result in subsequent tokens having a small proportion of attention score because of the causal mask; Calculating based on the sliding window may be difficult to obtain accurate global information.
> 3. The calculation of attention scores need to be done at every step, which requires additional computational and memory costs.
>
>
>
> Q:
>
> The "Method" section is incomplete, missing some important details. It is not clear how SinkQ identifies the sink tokens, which is a critical part of the proposed method. During inference, the tokens are ingested in a streaming manner, while the authors only describe how the outlier tokens can be identified after saving all keys in an offline manner on line 152.
>
>
>
> A:
>
> We further explain the quantization process here:
>
> We follow the quantization process of KIVI, performing quantization every G steps. At this point, we have G KV caches, and we directly calculate the magnitude of these key caches as the metric. Then, we select the outlier tokens based on this metric and quantize the remaining tokens.
>
> We have changed the related context in our paper (see Section 3.2).

---

> > ### Author Response · Authors · 2024-11-27
> > **part2**
> >
> > Q:
> >
> > Baselines are missing from the experiments. Since the authors adopt a mixed-precision quantization approach, mixed-precision baselines, such as KVQuant and ZipCache, are more relevant than the KIVI baseline.
> >
> >
> >
> > A:
> >
> > Thanks for your suggestion. We did not select these baselines mainly because there were some differences in settings which could lead to unfair comparisons. We have added the results of ZipCache, H2O, Streamingllm and SnapKV. The results are shown below and the experimental details can be found in Appendix B.
> >
> > | Llama2-7b-chat | Gsm8k(8)  | 8-cot     | 0-cot     | BBH(3)    | 3-cot     | 0-cot     | HE(p@1)   | p@10      | Avg       |
> > | -------------- | --------- | --------- | --------- | --------- | --------- | --------- | --------- | --------- | --------- |
> > | FP16           | 21.99     | 21.3      | 24.11     | 33.34     | 40.21     | 35.00     | 12.19     | 17.07     | 45.01     |
> > | KIVI           | 16.3      | 17.51     | 21.61     | 32.48     | 34.00     | 33.30     | 9.75      | 12.19     | 43.59     |
> > | Ours           | **19.86** | **19.33** | **22.52** | **33.33** | **34.43** | **33.74** | **11.58** | **14.63** | **43.75** |
> > | ZipCache       | 15.92     | 17.74     | 20.02     | 32.85     | 33.90     | 32.35     | 9.45      | 15.24     | 42.05     |
> >
> >
> >
> > | Llama2-7b-chat | Qasper    | GovReport | MultiNews | TREC      | TriviaQA  | SamSum   | LCC      | Repobench-P | Avg       |
> > | -------------- | --------- | --------- | --------- | --------- | --------- | -------- | -------- | ----------- | --------- |
> > | FP16           | 20.04     | 25.08     | 23.02     | 59.67     | 85.39     | 39.28    | 59.59    | 48.04       | 45.01     |
> > | KIVI           | **20.43** | 19.97     | 19.82     | 59.67     | **85.16** | 37.7     | 58.73    | 47.24       | 43.59     |
> > | SnapKV         | 18.96     | 18.73     | 19.64     | 59        | 84.84     | 38.22    | **60.5** | **50.08**   | 43.75     |
> > | H2O            | 17.51     | 18.85     | 19.88     | 50        | 84.22     | 38.09    | 58.23    | 49.66       | 42.05     |
> > | Streaming      | 15.31     | 19.39     | 18.99     | 51        | 83.11     | 36.8     | 57.57    | 47.33       | 41.19     |
> > | Ours           | 19.95     | **21.56** | **20.81** | **59.67** | 85        | **39.1** | 59.44    | 48.51       | **44.26** |
> >
> > We are also reproducing the results of KVQuant, but there are some issues with the results. We hope we can attach the results of KVQuant before the deadline.
> >
> >
> >
> >
> >
> > Q:
> >
> > The comparison with KIVI feels like an unfair comparison; since SinkQ directly adopts the KIVI approach (per-channel for keys and per-token for values, with residual cache) and adds sink tokens, the model quality will certainly outperform KIVI, but with memory and latency overheads. These overheads are not made very clear in Table 1 and 2.
> >
> >
> >
> > A:
> >
> > Our method aims to achieve high quantization accuracy improvement with minimal cost. In fact, we have indeed achieved this. Figure 3 shows the speed and memory performance of our method compared to KIVI and FP16 baseline.Our method is based on KIVI, but exploring improvement points on existing methods is a reasonable research strategy. So, this will not lead to unfair comparisons. However, it may be an unfair comparison between SinkQ and other methods like KVQuant, which have different settings (tuning&tuning-free, line 255).

---

### Official Review · Reviewer_tpM7 · 2024-11-04

**Soundness:** 3
**Presentation:** 3
**Contribution:** 2
**Rating:** 5
**Confidence:** 4

**Summary:**

This paper presents SinkQ, a 2-bit quantization technique designed to improve the memory efficiency of KV cache in LLMs. This method dynamically identifies and tracks "sink tokens"—tokens with high attention scores that would otherwise suffer from quantization errors—and excludes them from quantization to retain accuracy. The SinkQ approach offers notable memory and throughput benefits, achieving a $6.4\times$ reduction in memory usage and a $2.3\times$ increase in throughput, while minimally impacting model accuracy. It is implemented in a hardware-friendly way, providing compatibility with common LLM frameworks.

**Strengths:**

- This work provides thorough ablation studies and insightful findings on KV cache compression, particularly the role of attention distribution and the characteristics of tokens that function as attention "sinks."
- By excluding high-attention tokens from quantization, SinkQ effectively preserves accuracy under 2-bit quantization, making it feasible for real-time applications that require efficient memory usage.
- SinkQ’s compatibility with methods like KIVI, which uses channel-wise and token-wise quantization, demonstrates flexibility and potential for integration with other compression frameworks.

**Weaknesses:**

- The paper lacks an in-depth discussion of the computational overhead introduced by steps such as slicing, concatenation, and the calculation of outlier tokens. A profiling of these operations would help to understand the real-world efficiency impact.
- The paper compares SinkQ with KIVI but does not include comparisons with other KV cache compression methods, such as KVQuant, [1], GEAR [2] and SKVQ [3].
- SinkQ could be considered an innovation that combines aspects of token-dropping approaches, such as those in StreamingLLM [4] and H2O [5], with quantization strategies. The novelty may be seen as limited.

[1] Kvquant: Towards 10 million context length llm inference with kv cache quantization

[2] Gear: An efficient kv cache compression recipe for near-lossless generative inference of llm

[3] SKVQ: Sliding-window Key and Value Cache Quantization for Large Language Models

[4] Efficient streaming language models with attention sinks

[5] H2o: Heavy-hitter oracle for efficient generative inference of large language models

**Questions:**

- What is the overall overhead incured by additional operations, a profiling will be good.
- What is the process if the sink pool reaches its maximum capacity? Could you elaborate on the overflow outlined in lines 294-298?
- What are the performance metrics for SinkQ on models like Qwen and LongChat?

---

> ### Author Response · Authors · 2024-11-27
>
> Thank you for your valuable comments. We will address your concerns as follows.
>
> Q:
>
> The paper lacks an in-depth discussion of the computational overhead introduced by steps such as slicing, concatenation, and the calculation of outlier tokens. A profiling of these operations would help to understand the real-world efficiency impact.
>
> A:
>
> The computational overhead comes from two aspects.
>
> 1. In the compression stage, we calculate the magnitude of each token's key, perform threshold comparison, select the index, and quantify it. The cost of the sink operation is relatively high compared to quantization, but the compression is only performed every G steps (G is the group size in the paper and we set it to 128 in the main experiment), so this time cost can be almost negligible compared to the whole decoding process.
> 2. In the attention calculation stage, we need to calculate the qkv in the sink pool and cover the attention score according to the sink id, which has a certain cost.
>
> We take an example for illustration. We use SinkQ to infer an example which is randomly selected from GSM8K with an input length of 269 and an output length of 20. It has 2 compress steps and 20 decode steps. We record the time consumption in the attention block and create a pie chart in appendix C. The sink operation accounts for about 18% of the time. However, considering the pre-processing, post-processing and FFN calculation in the entire forward step, the time proportion of sink operations is also very small overall.
>
> We have also retained the profiler and added it to the supplementary material, which you can open with TensorBoard.
>
>
>
>
>
>
>
> Q:
>
> The paper compares SinkQ with KIVI but does not include comparisons with other KV cache compression methods, such as KVQuant, [1], GEAR [2] and SKVQ [3].
>
> A:
>
> Thanks for your suggestion. We did not select these baselines mainly because there were some differences in settings which could lead to unfair comparisons. For example, KVQuant uses many complex operations to improve accuracy and is not tuning-free; GRAR requires computation of low-rank matrix, which leads to additional latency and memory.
> To address your concerns, we have added the results of ZipCache, H2O, Streamingllm and SnapKV. The results are shown below and the experimental details can be found in Appendix B.
>
> | Llama2-7b-chat | Gsm8k(8)  | 8-cot     | 0-cot     | BBH(3)    | 3-cot     | 0-cot     | HE(p@1)   | p@10      | Avg       |
> | -------------- | --------- | --------- | --------- | --------- | --------- | --------- | --------- | --------- | --------- |
> | FP16           | 21.99     | 21.3      | 24.11     | 33.34     | 40.21     | 35.00     | 12.19     | 17.07     | 45.01     |
> | KIVI           | 16.3      | 17.51     | 21.61     | 32.48     | 34.00     | 33.30     | 9.75      | 12.19     | 43.59     |
> | Ours           | **19.86** | **19.33** | **22.52** | **33.33** | **34.43** | **33.74** | **11.58** | **14.63** | **43.75** |
> | ZipCache       | 15.92     | 17.74     | 20.02     | 32.85     | 33.90     | 32.35     | 9.45      | 15.24     | 42.05     |
>
>
>
> | Llama2-7b-chat | Qasper    | GovReport | MultiNews | TREC      | TriviaQA  | SamSum   | LCC      | Repobench-P | Avg       |
> | -------------- | --------- | --------- | --------- | --------- | --------- | -------- | -------- | ----------- | --------- |
> | FP16           | 20.04     | 25.08     | 23.02     | 59.67     | 85.39     | 39.28    | 59.59    | 48.04       | 45.01     |
> | KIVI           | **20.43** | 19.97     | 19.82     | 59.67     | **85.16** | 37.7     | 58.73    | 47.24       | 43.59     |
> | SnapKV         | 18.96     | 18.73     | 19.64     | 59        | 84.84     | 38.22    | **60.5** | **50.08**   | 43.75     |
> | H2O            | 17.51     | 18.85     | 19.88     | 50        | 84.22     | 38.09    | 58.23    | 49.66       | 42.05     |
> | Streaming      | 15.31     | 19.39     | 18.99     | 51        | 83.11     | 36.8     | 57.57    | 47.33       | 41.19     |
> | Ours           | 19.95     | **21.56** | **20.81** | **59.67** | 85        | **39.1** | 59.44    | 48.51       | **44.26** |
>
> We are also reproducing the results of KVQuant, but there are some issues with the results. We hope we can attach the results of KVQuant before the deadline.

---

> > ### Author Response · Authors · 2024-11-27
> > **part2**
> >
> > Q:
> >
> > SinkQ could be considered an innovation that combines aspects of token-dropping approaches, such as those in StreamingLLM [4] and H2O [5], with quantization strategies. The novelty may be seen as limited.
> >
> >
> >
> > A:
> >
> > SinkQ can indeed be seen as a combination of token eviction and KV Cache quantization, but its innovation lies elsewhere.
> >
> > 1. SinkQ reveals the outlier token phenomenon in channel-wise key quantization, which was not mentioned in previous work
> > 2. The identification method of the outlier token in SinkQ is different from other methods. It directly uses the magnitude of the key as the evaluation metric, which does not require the calculation and storage of the accumulated attention score at each step, but only needs to be calculated at the compression step. This greatly speeds up efficiency, and this method is perfectly adapted to our preliminary exploration.
> > 3. SinkQ only needs to retain a few outlier tokens to significantly increase quantization accuracy, while methods such as H2O require retaining more important tokens than ours.
> > 4. Compared to streamgllm, SinkQ has a more refined sink recognition scheme.
> >
> >
> >
> > For sink and heavy hitters, we provide further explanation:
> > We believe that attention sinks and heavy hitters are not completely identical concepts, and their differences are as follows:
> >
> > 1. Attention sinks refer to tokens with consistently high attention scores during the decoding process, with only a few tokens (usually less than 4).
> > 2. Heavy hitters are relatively important tokens in the decoding process compared to other tokens, typically accounting for over 10%-20% of all tokens.
> > 3. Heavy hitters contain attention sinks, but there are differences in the distribution within tokens and the number of tokens. We have plotted the accumulative attention scores (sorted) in Appendix D, indicating that attention sinks are more unique than heavy hitters.
> >
> >
> >
> > We further explain why our recognition strategy, namely the magnitude of the key, is better than using accumulative attention score:
> >
> > 1. Selecting outlier tokens based on the magnitude of the key can accurately identify them in the channel-wise key quantization, while the accumulative attention score may not (the top k of the attention score in Figure 1 (c) does not completely overlap with the top k of the magnitude of the key).
> > 2. The calculation of accumulative attention score may not be accurate under this setting. Calculating based on the global attention score may result in subsequent tokens having a small proportion of attention score because of the causal mask; Calculating based on the sliding window may be difficult to obtain accurate global information.
> > 3. The calculation of attention scores need to be done at every step, which requires additional computational and memory costs.
> >
> >
> >
> >
> >
> > Q:
> >
> > What is the process if the sink pool reaches its maximum capacity? Could you elaborate on the overflow outlined in lines 294-298?
> >
> >
> >
> > A:
> >
> > When the sink pool is full, we will perform a replacement. We add the new sink to the sink pool, while the discarded sink will be stored in another pool (in fact, it can be restored to the quantized KV Cache, but for the sake of simplicity, we choose to leave an extra pool with a size of 32). When the extra pool is full, we will no longer perform sink recognition because we have found that token exchange in the sink pool rarely occurs throughout the entire decode process.

---

### Note · Authors · 2024-12-15

I have read and agree with the venue's withdrawal policy on behalf of myself and my co-authors.